# BASED ON WHAT WE CAN CONTROL ARTIFICIAL NEURAL NETWORKS

## ABSTRACT

How can the stability and efficiency of Artificial Neural Networks (ANNs) be ensured through a systematic analysis method? This paper seeks to address that query. While numerous factors can influence the learning process of ANNs, utilizing knowledge from control systems allows us to analyze its system function and simulate system responses. Although the complexity of most ANNs is extremely high, we still can analyze each factor (e.g., optimiser, hyperparameters) by simulating their system response. This new method also can potentially benefit the development of new optimiser and learning system, especially when discerning which components adversely affect ANNs. Controlling ANNs can benefit from the design of optimiser and learning system, as (1) all optimisers act as controllers, (2) all learning systems operate as control systems with inputs and outputs, and (3) the optimiser should match the learning system. We will share the source code of this work after the paper has been accepted for publication.

## 1 INTRODUCTION

Controlling artificial neural networks (ANNs) has become an urgent issue on such a dramatically growing domain. Although ANN models, such as, vision models (e.g., CNN Krizhevsky et al. [2012], VGG19 Simonyan & Zisserman [2014], ResNet50 He et al. [2016a], EfficientNet Tan & Le [2019], ViT Dosovitskiy et al. [2020]), language models (e.g., BERT Devlin et al. [2018], GPT Radford et al. [2018], PaLM Chowdhery et al. [2022]), and generative models (e.g., GAN Goodfellow et al. [2014], VAE Kingma & Welling [2013], Stable Diffusion Models Ho et al. [2020]; Rombach et al. [2022]), all require input and output, as they aim to map the gap between their output and the desired output. However, basically, CNN-based vision models prefer SGDM Qian [1999] optimiser, and generative models tend to rely on AdaM optimiser. Using various architecture on CNN-based vision models (e.g., from VGG19 to ResNet50, from GAN to CycleGAN Zhu et al. [2017], and from CNN to FFNN Hinton [2022]) yield significantly varied results for classification and generation tasks. Two critical questions arise: **(1)** why some of them satisfy the corresponding optimiser, **(2)** based on what to propose an advanced ANN architecture and a proper optimiser.

Compared to existing era-acrossing optimisers, such as SGD Robbins & Monro [1951]; Cotter et al. [2011]; Zhou & Cong [2017], SGDM Qian [1999]; Liu et al. [2020], AdaM Kingma & Ba [2014]; Bock et al. [2018], PID Wang et al. [2020], and Gaussian LPF-SGD Bisla et al. [2022], we proposed a FuzzyPID optimiser modified by fuzzy logic to avoid vibration during PID optimiser learning process. Referring to Gaussian LPF-SGD (GLFP-SGD), we also proposed two filter processed SGD methods according to the low and high frequency part during the SGD optimiser learning process: low-pass-filter SGD (LPF-SGD) and high-pass-filter SGD (HPF-SGD). To achieve stable and convergent performance, we simulate these above optimisers on the system response to analyze their attributes. When using simple and straightforward architecture (without high techniques, such as, BN Ioffe & Szegedy [2015], ReLU Nair & Hinton [2010], pooling Wu & Gu [2015], and exponential or cosine decay Li et al. [2021]), we found their one step system response are always consistent with their training process. Therefore, we conclude that every optimiser actually can be considered as a controller that optimise the training process. Results using HPF-SGD indicate that the high frequency part using SGD optimiser significantly benefits the learning process and the classification performance.

To analyze the learning progress of most ANNs, for example, CNN using backpropagation algorithm, FFNN using forward-forward algorithm, and GAN such a generative model using random noise to generate samples. We assume above three mentioned models here essentially can be represented by corresponding control systems. But the difficulty is that when using different optimisers, especially, AdaM, we cannot analyze its stability and convergence, as the complexity is extremely high. Thus, we use MATLAB Simulink to analyze their system response, as well as their generating response. Experiment results indicate that advanced architectures and designs of these three ANNs can improve the learning, such as residual connections (RSs) on ResNets, a higher Threshold on FFNN, and a cycle loss function on CycleGAN.

Based on the knowledge of control systems Nise [2020], designing proper optimisers (or controllers) and advanced learning systems can benefit the learning process and complete relevant tasks (e.g., classification and generation). In this paper, we design two advanced optimisers and analyze three learning systems relying on the control system knowledge. The contributions are as follows:

**Optimisers are controllers.** **(1)** PID and SGDM (PI controller) optimiser performs more stable than SGD (P controller), SGDM (PI controller), AdaM and fuzzyPID optimisers on most residual connection used CNN models. **(2)** HPF-SGD outperforms SGD and LPF-SGD, which indicates that high frequency part is significant during SGD learning process. **(3)** AdaM is an adaptive filter that combines an adaptive filter and an accumulation adaptive part.

**Learning systems of most ANNs are control systems.** **(1)** Most ANNs present perfect consistent performance with their system response. **(2)** We can use proper optimisers to control and improve the learning process of most ANNs.

**The Optimiser should match the learning system.** **(1)** RSs based vision models prefer SGDM, PID and fuzzyPID optimisers. **(2)** RS mechanism is similar to AdaM. particularly, SGDM optimizes the weight of models on the time dimension, and RS optimizes the model on the space dimension. **(3)** AdaM significantly benefits FFNN and GAN, but PID and FuzzyPID dotes CycleGAN most.

## 2 PROBLEM STATEMENT AND PRELIMINARIES

To make ANNs more effective and adaptive to specific tasks, controlling ANNs has become necessary. We initialize a parameter of a node in the ANN model as a scalar $\theta_0$. After enough time of updates, the optimal value of $\theta^*$ can be obtained. We simplify the parameter update in ANN optimisation as a one-step response (from $\theta_0$ to $\theta^*$) in the control system. The Laplace transform of $\theta^*$ is $\theta^*/s$. We denote the weight $\theta(t)$ at iteration $t$. The Laplace transform of $\theta(t)$ is denoted as $\theta(s)$, and that of error $e(t) = \theta^* - \theta(t)$ as $E(s)$:

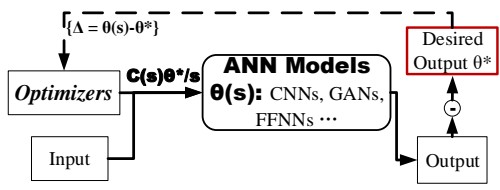

Figure 1: The schematic structure of training ANN models. C(s) is the controller to train the target ANN model.

$$E(s) = \frac{\theta^*}{s} - \theta(s) \tag{1}$$

Considering the collaboration of backward and forward algorithms, the Laplace transform of the training process is

$$U(s) = (Controller1 + Controller2 \cdot F(s)) \cdot E(s) \tag{2}$$

$F(s)$ is the forward system which has the capability to affect $U(s)$ beforehand. In our case, $u(t)$ corresponds to the update of $\theta(t)$. $Controller1$ is the parameter update algorithm for the backward process, and $Controller2$ is the parameter update algorithm for the forward process. Therefore, we replace $U(s)$ with $\theta(s)$ and $E(s)$ with $(\theta^*/s) - \theta(s)$. Equation 2 can be rewritten as

$$\theta(s) = (Controller1 + Controller2 \cdot F(s)) \cdot \left(\frac{\theta^*}{s} - \theta(s)\right) \tag{3}$$

Finally, we simplify the formula of training a model as:

$$\theta(s) = \frac{Controller}{Controller + 1} \cdot \frac{\theta^*}{s} \tag{4}$$

where $Controller = Controller1 + Controller2 \cdot F(s)$. $\theta^*$ denotes the optimal model which we should get at the end. Simplifying $\theta(s)$ further as below:

$$\theta(s) = \mathbf{Controller(s)} \cdot \mathbf{C(s)} \tag{5}$$

where $\mathbf{Controller(s)} = Controller/(Controller+1)$, and $\mathbf{C(s)} = \theta^*/s$. Based on above analytic thought, as shown in Figure 1 there are two ways to obtain an optimal $\theta(s)$ and to make the training process better: **(1)** using a better **Controller** and **(2)** constructing a better training or control system $\mathbf{C(s)}$.

## 3 OPTIMISERS ARE CONTROLLERS

In this section, we review several widely used optimisers, such as SGD Robbins & Monro [1951]; Cotter et al. [2011]; Zhou & Cong [2017], SGDM Qian [1999]; Liu et al. [2020], AdaM Kingma & Ba [2014]; Bock et al. [2018], PID-optimiser Wang et al. [2020] and Gaussian LPF-SGD Bisla et al. [2022]. In the training process of most ANNs, there are diverse architectures used to satisfy various tasks. We analyze the performance of optimisers in terms of one node of backpropagation based ANN models. Please see the proof in Appendix A.

### 3.1 ADAM OPTIMISER

AdaM Kingma & Ba [2014] has been used to optimise the learning process of most ANNs, such as GAN, VAE, Transformer-based models, and their variants. We simplify the learning system of using AdaM on ANNs as below:

$$\theta(s) = \frac{K_p s + K_i}{M s^2 + (K_p - M ln\beta_1)s + K_i} \cdot \frac{\theta^*}{s} \tag{6}$$

where $M$ is an adaption factor which will dynamically adjust the learning during the training process, and it can be derived from:

$$M = \frac{1}{\sqrt{\frac{\sum_{i=0}^{t} \beta_2^{t-i}(\partial L_t/\partial \theta_t)^2}{\sum_{i=0}^{t} \beta_2^{i-1}} + \epsilon}} \cdot \frac{1}{\sum_{i=0}^{t} \beta_1^{i-1}} \tag{7}$$

Apart from the adaption part $M$, AdaM can be thought as the combination of SGDM and an adaptive filter with the cutoff frequency $\omega_c = ln(\beta_1)$.

### 3.2 FILTER PROCESSED SGD OPTIMISER

SGD learning process can be filtered under carefully designed filters. GLPF-SGD Bisla et al. [2022] used a low pass Gaussian-filter to smooth the training process, as well as actively searching the flat regions in the Deep Learning (DL) optimisation landscape. Eventually, we simplify the learning system of using SGD with filters on ANNs as below:

$$\theta(s) = \frac{Gain \cdot \prod_{i=0}^{m} (s + h_i)}{Gain \cdot \prod_{i=0}^{m} (s + h_i) + \prod_{j=0}^{n} (s + l_j)} \cdot \frac{\theta^*}{s} \tag{8}$$

where designed $Filter$ have the order, such as $n$ for the low pass and $m$ for the high pass ($h_i$ is the coefficient of the high pass part and $l_i$ is the coefficient of the low pass part), and $Gain$ is the gain factor:

$$Filter = Gain \cdot \frac{(s + h_0)(s + h_1)...(s + h_m)}{(s + l_0)(s + l_1)...(s + l_n)} \tag{9}$$

### 3.3 PID AND FUZZYPID OPTIMISER

Based on PID optimiser Wang et al. [2020], we design a PID controller which is optimised by fuzzy logic to make the training process more stable while keeping the dominant attribute of models. For instance, the ability to resist the disturbance of the poisoned samples, the quick convergent speed and the competitive performance.

There are two key factors which affect the performance of the Fuzzy PID optimiser: (1) the selection of Fuzzy Universe Range $[-\varphi, \varphi]$ and (2) Membership Function Type $f_m$.

$$\widehat{K}_{\text{P,I,D}} = K_{\text{P,I,D}} + \Delta K_{\text{P,I,D}} \tag{10}$$

$$\Delta K_{\text{P,I,D}} = Defuzzy(E(s), Ec(s)) \cdot K_{\text{P,I,D}}$$
$$Defuzzy(s) = f_m(round(-\varphi, \varphi, s)) \tag{11}$$

where $\Delta K_{\text{P,I,D}}$ refer to the default gain coefficients of $K_{\text{P}}$, $K_{\text{I}}$ and $K_{\text{D}}$ before modification. $E(s)$ is the back error, and $Ec(s)$ is the difference product between the $Laplace$ of $e(t)$ and $e(t-1)$. The Laplace function of this model $\theta(s)$ eventually becomes:

$$\theta(s) = \frac{\widehat{K}_d s^2 + \widehat{K}_p s + \widehat{K}_i}{\widehat{K}_d s^2 + (\widehat{K}_p + 1)s + \widehat{K}_i} \cdot \frac{\theta^*}{s} \tag{12}$$

where $\widehat{K}_p$, $\widehat{K}_i$ and $\widehat{K}_d$ should be processed under the fuzzy logic. By carefully selecting the learning rate $r$, $\theta(s)$ becomes a stable system.

The PID Ang et al. [2005] and Fuzzy PID Tang et al. [2001] controllers have been used to control a feedback system by exploiting the present, past, and future information of prediction error. The advantages of a fuzzy PID controller includes that it can provide different response levels to non-linear variations in a system. At the same time, the fuzzy PID controller can function as well as a standard PID controller in a system where variation is predictable.

## 4 CONTROL SYSTEMS OF ANNs

In this section, to systematically analyze the learning process of ANNs, we introduce three main common-used control systems that we believe can be respectively connected to backpropagation based CNNs, forward-forward algorithm based FFNNs, and GANs: **(1)** backward control system, **(2)** forward control system using different hyperparameters, and **(3)** backward-forward control system on different optimisers and hyperparameters. Please see the proof in Appendix B.

### 4.1 BACKWARD CONTROL SYSTEM

Traditional CNNs use the backpropagation algorithm to update initialized weights, and based on errors or minibatched errors between real labels and predicted results, optimisers are used to control on how the weight should be updated. According to the deduction of PID optimiser Wang et al. [2020], the training process of Deep Neural Networks (DNNs) can be conducted under a step response of control systems. However, most common-used optimisers have their limitations, such as **(1)** SGD costs a very long term to reach convergence, **(2)** SGDM also has the side effect of long term convergence even with the momentum accelerating the training, **(3)** AdaM presents a frequent vibration during the training because of the merging of momentum and root mean squared propagation (RM-Sprop), **(4)** PID optimiser has better stability and convergence speed, but the training process is still vibrating. This proposed fuzzyPID optimiser can keep the learning process more stable, because it can be weighted towards types of responses, which seems like an adaptive gain setting on a standard PID optimiser. Finally, we get the system function $\theta(s)$ of ANNs by using FuzzyPID optimisers as an example below:

$$\theta(s) = \frac{FuzzyPID}{FuzzyPID + 1} \cdot \frac{\theta^*}{s} \tag{13}$$

### 4.2 FORWARD-FORWARD CONTROL SYSTEM

The using of forward-forward computing algorithm was systematically analyzed in forward-forward neural network Hinton [2022] which aims to track features and figure out how ANNs can extract them from the training data. The Forward-Forward algorithm is a greedy multilayer learning procedure inspired by Boltzmann machines Hinton et al. [1986] and noisy contrastive estimation Gutmann & Hyvärinen [2010]. To replace the forward-backward passes of backpropagation with two forward passes that operate on each other in exactly the same way, but on different data with opposite goals. In this system, the positive pass operates on the real data and adjusts the weights to increase the goodness in each hidden layer; the negative pass operates on the negative data and adjusts the weights to reduce the goodness in each hidden layer. According to the training process of FFNN, we get its system function $\theta(s)$ as below:

$$\theta(s) = \left\{ \left( -(1-\lambda)\frac{\theta^*}{s} + \lambda\frac{\theta^*}{s} - \left[ \theta(s) - \frac{Th}{s} \right] \right) \right\} \cdot Controller \tag{14}$$

where $\lambda \in [0,1]$ is the portion of positive samples, and $Th$ is the given Threshold according to the design Hinton [2022]. Input should contain negative and positive samples, and by adjusting the Threshold $Th$, the embedding space can be optimised. In each layer, weights should be updated on only corresponding errors that can be computed by subtracting the Threshold $Th$. We finally simplify $\theta(s)$ as:

$$\theta(s) = \frac{1}{Controller + 1} \cdot \left( \frac{(2\lambda - 1)\theta^* + Th}{s} \right) \tag{15}$$

Because $(2\lambda - 1)\theta^* + Th \geq 0$, the system of FFNN is stable. Additionally, when $\lambda = 0.5$ and $Th = 1.0$, the learning system of FFNN (the second half part of Equation 15) will become to that of backpropagation based CNN, as we assume $\theta^* \approx 1.0$. When $\lambda = 0.5$, the optimal result $\theta^*$ has no relationship with the learning system.

### 4.3 BACKWARD-FORWARD CONTROL SYSTEM

GAN is designed to generate samples from the Gaussian noise. The performance of the GAN depends on its architecture Zhou et al. [2023]. The generative network uses random inputs to generate samples, and the discriminative network aims to classify whether the generated sample can be classified Goodfellow et al. [2014]. We get its $\theta(s)$ as below:

$$\theta_D(s) = controller \cdot \theta_G(s) \cdot E(s) \tag{16}$$

$$\theta_G(s) = controller \cdot E(s) \tag{17}$$

$$E(s) = \frac{\theta_D^*}{s} - \theta_D(s) \tag{18}$$

where $\theta_D(s)$ is the desired Discriminator, $\theta_G(s)$ is the desired Generator. $E(s)$ is the feed-back error. $\theta_G^*$ is the optimal solution of the generator, and $\theta_D^*$ is the optimal solution of the discriminator.

Eventually, we simplify $\theta_G(s)$ and $\theta_D(s)$ as below:

$$\theta_G(s) = \frac{1}{2} \cdot \left( \frac{\theta_D^*}{Controller} \pm \sqrt{(\frac{\theta_D^*}{Controller})^2 - \frac{4}{s}} \right) \tag{19}$$

$$\theta_D(s) = \theta_G^2(s) \tag{20}$$

where if set $\theta_G(s) = 0$, we get one pole point $s = 0$. When using SGD as the *controller*, $\theta_G(s)$ is a marginally stable system.

## 5 EXPERIMENTS

### 5.1 SIMULATION

As we believe that the training process of most ANNs can be modeled as the source response of control systems, we use Simulink (MATLAB R2022a) to simulate their response to different sources. For the classification task, because all models aim to classify different categories, we set a step source as illustrated in Wang et al. [2020]. For the sample generation task, to get a clear generating result, we use a sinusoidal source.

## 5.2 EXPERIMENT SETTINGS

We train our models on the MNIST LeCun et al. [1998], CIFAR10 Krizhevsky et al. [2009], CI-FAR100 Krizhevsky et al. [2009] and TinyImageNet Le & Yang [2015] datasets. For an apple-to-apple comparison, our training strategy is mostly adopted from PID optimiser Wang et al. [2020] and FFNN Hinton [2022]. To optimise the learning process, we **(1)** firstly use seven optimisers for the classification task on backpropagation algorithm based ANNs. **(2)** Secondly, we choose some important hyperparameters and simulate the learning process of FFNN. **(3)** Lastly, to improve the stability and convergence during the training of GAN, we analyze its system response on various optimisers. All models are trained on single Tesla V100 GPU. All the hyper-parameters are presented in Table 3 of Appendix E.

### 5.2.1 BACKWARD CONTROL SYSTEM

We design one neural network using backpropagation algorithm with 2 hidden layers, setting the learning rate $r$ at 0.02 and the fuzzy universe range $\varphi$ at $[-0.02, 0.02]$. We initialize $K_P$ as 1, $K_I$ as 5, and $K_D$ as 100. Thus, we compare seven different optimisers: SGD (P controller), SGDM (PI controller), AdaM (PI controller with an Adaptive Filter), PID (PID controller), LPF-SGD, HPF-SGD and FuzzyPID (fuzzy PID controller) on the above ANN model. We set Gaussian membership function as the default membership function. See filter coefficients in Table 4 of Appendix E. In Table 5 of Appendix E, there is a set of hyperparameters that we have used to trian CIFAR10, CIFAR100 and TinyImageNet.

### 5.2.2 FORWARD-FORWARD CONTROL SYSTEM

Following the forward-forward algorithm Hinton [2022], we design one forward-forward neural network (FFNN) with 4 hidden layers each containing 2000 ReLUs and full connectivity between layers, by simultaneously feeding positive and negative samples into the model to teach it to distinguish the handwriting number (MNIST). We also carefully select the proportion of positive and negative samples. The length of every block is 60.

### 5.2.3 BACKWARD-FORWARD CONTROL SYSTEM

To demonstrate the relationship between the control system and the learning process of some complex ANNs, we choose the classical GAN Goodfellow et al. [2014]. Both the generator and the discriminator comprise 4 hidden layers. To verify the influence of different optimisers on GAN, we employ SGD, SGDM, AdaM, PID, LPF-SGD, HPF-SGD and fuzzyPID to generate the handwriting number (MNIST). We set the learning rate at 0.0002 and the total number of epochs at 200.

## 6 RESULTS AND ANALYSIS

In this section, we present simulation performance, classification accuracy, error rate and generation result, using different optimisers and advanced control systems.

### 6.1 BACKWARD CONTROL SYSTEM ON CNN

Table 1: The results of ANN based on the backpropogation algorithm on MNIST data. Using the 10-fold cross-validation, the average and standard variance results are shown below.

| optimiser | SGD | SGDM | Adam | PID | LPF-SGD | HPF-SGD | FuzzyPID |
|---|---|---|---|---|---|---|---|
| **Training** *Accuracy* | $91.48_{\pm0.03}$ | $97.78_{\pm0.00}$ | $99.46_{\pm0.02}$ | $99.45_{\pm0.01}$ | $11.03_{\pm0.01}$ | $93.35_{\pm0.02}$ | $99.73_{\pm0.09}$ |
| **Testing** *Accuracy* | $91.98_{\pm0.05}$ | $97.11_{\pm0.02}$ | $97.81_{\pm0.10}$ | $98.18_{\pm0.02}$ | $10.51_{\pm0.03}$ | $93.45_{\pm0.09}$ | $98.24_{\pm0.10}$ |

Before doing the classification task, we firstly simulate the step response of backpropagation based ANNs on each controller (optimiser). As observed in Figure 2b and Figure 2c, AdaM optimiser can rapidly converge to the optimal but with an obvious vibration. Although FuzzyPID cannot rapidly converge to the optimal, there is no obvious vibration during the training. Other optimisers, such as HPF-SGD, SGDM and PID, perform lower than AdaM and FuzzyPID in terms of the training

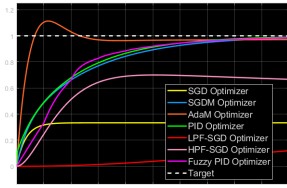 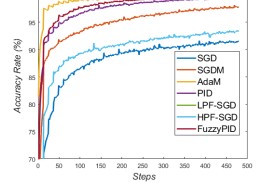 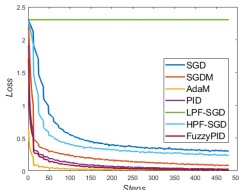

(a) The step response of CNN using different controllers (or optimisers).

(b) The training curve of CNN using different optimisers (or controllers) on MNIST.

(c) The loss curve of CNN using different optimisers (or controllers) on MNIST.

Figure 2: The step response, training curve and loss curve using different controllers, such as SGD, SGDM, AdaM, PID, LPF-SGD, HPF-SGD and FuzzyPID optimisers.

process. In Figure 2a, the response of AdaM controller is faster than others, and FuzzyPID follows it. However, due to the overshoot on AdaM, the stability of ANN system when using the AdaM controller tends to be lower. This overshoot phenomenon is reflected on the training process of Adam optimising in Figure 2b and Figure 2c.

We summarize the result of classifying MNIST in Table 1. Under the same condition, SGD optimiser reaches the testing accuracy at 91.98%, but other optimisers can reach above 97%. FuzzyPID gets the highest training and testing accuracy rates using Guassian membership function. In Figure 2, if considering the rise time, the settling time and the overshoot, the fuzzy optimiser outperforms other optimisers. A better optimiser (or controller) that has inherited advanced knowledge and sometimes has been effectively designed is beneficial for the classification performance.

## 6.2 FORWARD FORWARD CONTROL SYSTEM ON FFNN

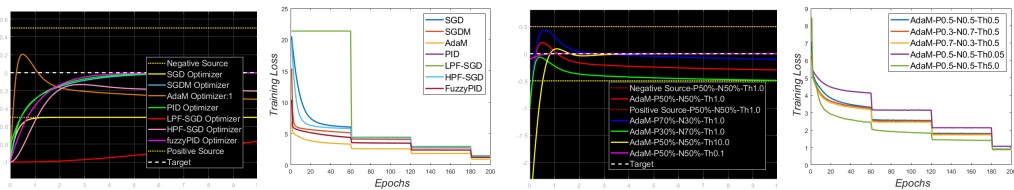

(a) The system response of FFNN on corresponding optimisers.

(b) The loss curve of FFNN on corresponding optimisers.

(c) The system response of FFNN on corresponding hyperparameters.

(d) The loss curve of FFNN on corresponding hyperparameters.

Figure 3: The step response and loss curve of FFNN using different controllers and various hyperparameters.

Table 2: The error rate (%) of FFNN using different optimisers and various hyperparameters on MNIST. Using the 10-fold cross-validation, the average and standard variance results are shown below.

| Method | 50% P, 50% N, Th=1.0, SGD | 50% P, 50% N, Th=1.0, SGDM | 50% P, 50% N, Th=1.0, Adam | 50% P, 50% N, Th=1.0, PID | 50% P, 50% N, Th=1.0, LPF-SGD | 50% P, 50% N, Th=1.0, HPF-SGD | 50% P, 50% N, Th=1.0, FuzzyPID | 30% P, 50% N, Th=1.0, Adam | 70% P, 30% N, Th=1.0, Adam | 50% P, 50% N, Th=0.1, Adam | 50% P, 50% N, Th=10.0, Adam |
|---|---|---|---|---|---|---|---|---|---|---|---|
| **Train Error** | 68.96 ±0.79 | 24.66 ±0.23 | 4.57 ±0.23 | 14.90 ±0.11 | 93.00 ±0.15 | 48.24 ±0.14 | 14.96 ±0.19 | 4.82 ±0.09 | 3.61 ±0.08 | 6.44 ±0.10 | 1.15 ±0.05 |
| **Test Error** | 68.85 ±0.90 | 24.02 ±0.30 | 5.00 ±0.30 | 14.38 ±0.15 | 92.89 ±0.36 | 48.45 ±0.23 | 14.43 ±0.25 | 5.31 ±0.13 | 4.37 ±0.11 | 6.52 ±0.13 | 1.35 ±0.08 |

We also simulate the control system of this proposed FFNN and compare its system response on different hyperparameters. In Figure 3, SGD controller still cannot reach the target, and AdaM controller reacts fastest approaching to the target. However, SGDM controller lags behind PID in terms of the step response. Because of the low frequency part of LPF-SGD, it climbs slower than

HPF-SGD. Although the differential coefficient D of PID optimiser can help reduce overshoot and overcome oscillation and reduce the adjustment time, its performance cannot catch up with AdaM. Compared to Table 2, AdaM outperforms other optimisers in terms of error rates, and the performance of these seven optimisers are echoing Figure 3a. A higher portion of positive samples can contribute to the classification, and a higher $Threshold$ can benefit more. For the step response in Figure 3c, although AdaM ($Threshold = 0.5$, $portion$ of positive samples is 70%, and $portion$ of negative samples is 30%) and AdaM ($Threshold = 0.5$, $portion$ of positive samples is 50%, and $portion$ of negative samples is 50%) rise fatest, the final results in Table 2 present that AdaM ($Threshold = 5.0$, $portion$ of positive samples is 50%, and $portion$ of negative samples is 50%) get a lower error rate.

## 6.3 BACKWARD-FORWARD CONTROL SYSTEM ON GAN

For the sample generation task, we also simulate the system response of GANs on each controllers (optimisers) and summarize the result in Figure 5. Apart from AdaM, LPF-SGD and HPF-SGD, all controllers have obvious noise, and interestingly, this phenomenon can be seen in Figure 4. The generated MNIST using Adam optimiser has no noise and can be easily recognized, and not surprised, the source response of AdaM in Figure 5 can finally converge. Figure 4 and Figure 5 mutually echo each other. Eventually, when using classical GAN to generate samples, AdaM should be the best optimiser to optimise the update of weights. The generated MNIST sample sometimes cannot be recognized, and GAN generates only same samples. One reason for this can be observed in Figure 5, where the sinusoidal signals generated by

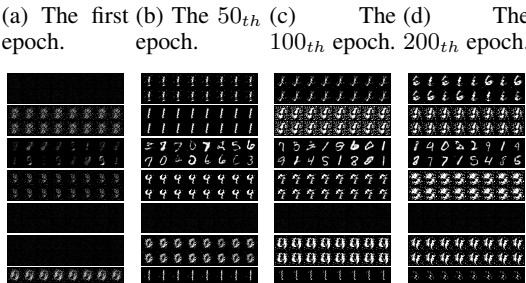

(a) The first epoch. (b) The $50_{th}$ epoch. (c) The $100_{th}$ epoch. (d) The $200_{th}$ epoch.

Figure 4: The generated samples from classical GAN on corresponding optimisers (from top to bottom is respectively SGD, SGDM, AdaM, PID, LPF-SGD, HPF-SGD, and FuzzyPID).

these four controllers, such as PID, LPF-SGD, HPF-SGD and FuzzyPID move up and down, potentially leading to an unstable and same generation output.

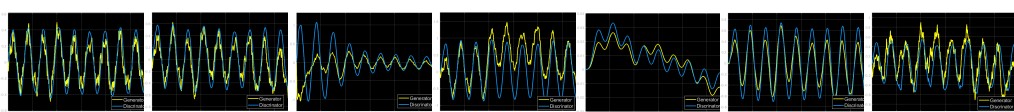

Figure 5: The system response of Classical GAN on different hyperparameters and various optimisers. Optimiser from left to right is respectively SGD, SGDM, AdaM, PID, LPFSGD, HPFSGD, and FuzzyPID. (Blue is the discriminator, and yellow is the generator)

## 7 DISCUSSION

### 7.1 WHY VARIOUS OPTIMISERS ARE CONTROLLERS DURING THE LEARNING PROCESS?

Under the same training condition (e.g., same architecture and hyperparameters), corresponding optimisers can tackle with specific tasks. Residual connection used vision models prefer SGDM, HPF-SGD and PID optimisers (Seen from Figure 14 of Appendix F). There is an obvious overshoot on the step response of AdaM controller (Seen from Figure 10), and a similar vibration can be found in the testing curve of Figure 14 of Appendix F. The classification task always needs a rapid response to save learning resources, but if stability and robustness are the priorities, we should set others as the opimizer, such as PID or FuzzyPID optimiser, which under fuzzy logic adjustment, demonstrates a superior step response (can be seen from Figure 2a). Moreover, for the generation task, GAN satisfies AdaM optimiser. We found that the adaptive part of AdaM can rapidly adjust the learning process. However, other optimisers, such as SGD, SGDM and PID, generate samples with

obvious noise and output the same samples make the generated sample cannot be recognized easily (can be seen from Figure 4 and Figure 5). For particular needs (e.g., Image-to-Image Translation), CycelGAN, this advanced generation system was proposed to generate samples from one data pool and to improve its domain adaption on the target data pool. Coincidentally, we found that CycleGAN has a preference for the PID optimiser. Therefore, it is necessary to design a stable and task-satisfied optimiser on a specifically designed learning system. However, given that the system functions of most learning systems are extremely complex, simulating their system responses has become a viable way to analyze them. We conclude that to achieve best performance, every ANN should use the proper optimiser according to its learning system.

## 7.2 How various learning systems can be analyzed?

Numerous advanced components have enhanced ANNs. Conducting a quantitative analysis on each of them can pave the way for the development of new optimisers and learning systems. For the classification task using a backward control system, in one node of the learning system, and in terms of analyzing a single component, the rise time, peak time, overshoot (vibration), and settling time Wang et al. [2020]; Nise [2020] can be the metrics to evaluate the performance of such component on learning systems. To visualize the learning process, FFNN was proposed by Hinton [2022] , and effectively, this forward-forward-based training system also can achieve competitive performance compared to backpropagation-based models. The $Threshold$ – one hyperparameter – can significantly benefit the convergence speed, as it has the effect of proportional adjustment (same as a stronger P in PID controller). The portion of positive samples can slightly affect the classification result, as because the proportional adjustment is too weak on FFNN learning system (Seen from Equation 15). Additionally, the system response on various sources can also serve as a metric to evaluate the learning system. We conclude that there are two main branches to improve ANNs: **(1)** develop a proper optimiser; **(2)** design a better learning system. On the one hand, for example, the system response of GAN has high-frequency noise and cannot converge using SGD, SGDM and PID optimisers (seen from Figure 5). One possible solution is adding an adaptive filter. Thus, AdaM outperforms other optimisers on generating samples (Seen from Figure 4). The overshoot of AdaM and SGDM during the learning process of classification tasks can accelerate the convergence, but its side-effect of vibration brings us to PID and FuzzyPID. Therefore, developing a task-matched optimiser according to the system response determines the final performance of ANNs. On the other hand, to satisfy various task requirements, learning systems also should become stable and fast. For example, $\theta_G(s)$ has two system functions as derived from Eq 19), to offset the side effect by considering the possible way using extra generator. That can explain why other advanced GANs using multi-generators (e.g., CycleGAN) can generate high-quality samples than the classical GAN.

## 8 Limitations

Although we systematically proved that **(1)** the optimiser acts as a controller and **(2)** the learning system functions as a control system, in this preliminary work, there are three obvious limitations: **a.** we cannot analyze larger models due to the complexity introduced by advanced techniques; **b.** the system response of some ANNs (e.g., FFNN) may not perfectly align with their real performance; **c.** we cannot always derive the solution of complex learning system.

## 9 Conclusion

In this study, we showed comprehensive empirical study investigating the connection between control systems and various learning systems of ANNs. We provided a systematic analysis method for several ANNs, such as CNN, FFNN, GAN, CycleGAN, and ResNet on several optimisers: SGD, SGDM, AdaM, PID, LPF-SGD, HPF-SGD and FuzzyPID. By analyzing the system response of ANNs, we explained the rationale behind choosing appropriate optimisers for different ANNs. Moreover, designing better learning systems under the use of proper optimiser can satisfy task requirements. In our future work, we will intend to delve into the the control system of other ANNs, such as Variational Autoencoders (VAEs), diffusion models, Transformer-based models and so on, aw well as the development of optimisers, as we believe the principles of control systems can guide improvements in all ANNs and optimisers.

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
