# OpenReview forum: "Based on What We Can Control Artificial Neural Networks"
_ICLR.cc/2024/Conference — Submitted to ICLR 2024_

### Official Review · Reviewer_egTB · 2023-10-15

**Soundness:** 1 poor
**Presentation:** 1 poor
**Contribution:** 2 fair
**Rating:** 3
**Confidence:** 3

**Summary:**

This paper presents a control system perspective of the ANNs training process and views the training algorithms as controllers. In addition, the authors analyzed several optimization methods (SGD, SGDm, AdaM, PID, LPF-SGD, HPF-SGD, Fuzzy PID) on different types of ANNs (CNN, FFNN, GAN, CycleGAN, ResNet). Through experimental studies, the authors conclude that different ANNs should use different training algorithms to achieve their best performance.

**Strengths:**

The control system perspective on the ANN training process is nice, although it is not new. Combining PID controller and fuzzy logic is novel to the best of my knowledge. There is a potential that the Fuzzy PID controller can actually improve the training process.

**Weaknesses:**

The presentation quality of the current manuscript can be further improved.

The novelty of this paper is weak. The idea of interpreting ANN training as controller design is not new. The idea of Fuzzy PID is new but incremental compared to [Wang2020]. In addition, the authors did not show the advantage of Fuzzy PID compared to other optimizers in the numerical section. It would be great if the authors could provide a rigorous proof of the convergence of the training algorithms through the lens of stability theory in control literature.

The dataset used in the numerical section is limited, and the model used in this part is too simple and unrepresentative. The results obtained in the main paper are inconsistent with those in the appendix (Figure 2 vs. Figure 15).

**Questions:**

1. Eq 2: it is abnormal to have 'text' in an equation. Many variables in Eq 7 are not well introduced.
2. In section 2, what is theta_0? is it the weight of a specific node? Do you only consider one node here?
3. Eq 1: how can we know the value  theta^* ahead? More details on the derivation of eq 1 to 5 is needed.
4. In Figure 1, what is the output? is it the output of the ANN model of the model weight? This figure is a bit confusing.
5. Under eq 15, why the system FFNN is stable under that inequality condition? Could you please explain more?
6. In Table 1, the results for LPF-SGD are strangely low, any intuition?
7. From Fig 2(b) and 2(c), AdaM clearly is the best one, why the stability of AdaM is bad? Also, the expression "stability is a lower" does not make sense from a control perspective, you either say stable or unstable.
8. From Fig 14-15, it seems that SGD and SGD-m works better than Fuzzy PID, PID, and AdaM for larger dataset and other models. This is very different than the case for MINIST results as shown in Fig 2, why is this? Any intuition?

---

### Official Review · Reviewer_jZKP · 2023-10-30

**Soundness:** 2 fair
**Presentation:** 2 fair
**Contribution:** 1 poor
**Rating:** 1
**Confidence:** 4

**Summary:**

This paper interprets the training of neural networks as a feedback controller on the parameter of the network. Several commonly used training algorithms are interpreted as proportional or PI controllers.

**Strengths:**

+ Neural network training, or understanding why neural network training works is an interesting question.
+ The interpretation in the paper is interesting and uses control theoretic ideas that are well-known.

**Weaknesses:**

- The technical problem considered in the paper is too simple to represent training of neural networks in my opinion. In particular, it assumes there is a single equilibrium, $\theta^*$, and the feedback is in the error between $\theta$ and $\theta^*$. A better model would be that we have the output $y(\theta)$, for example, the loss on the training set. But there are many $\theta$'s that give the same loss. For example, suppose that we can make the training loss 0, and $y(\theta_1*)=y(\theta_2*)=...=y(\theta_n^*)=0$ for some $n$. The challenge is that $n$ maybe very large, we don't know any of the $\theta_i^*$, and the feedback is on the error in $y$. The system is actually nonlinear, with many equilibriums, and convergence is much harder to understand.

**Questions:**

Looking at some nonlinearities would help. For example, just having one layer of ReLU, and see if the theory in the paper still works.

---

### Official Review · Reviewer_WG3Y · 2023-11-04

**Soundness:** 1 poor
**Presentation:** 1 poor
**Contribution:** 2 fair
**Rating:** 1
**Confidence:** 2

**Summary:**

The paper investigates the point of view of considering optimization neural networks as a control problem. They then investigate different known optimizers and controllers on a set of tasks.  I had great difficulty understanding the logic of this paper, my best guess is that the goal is to understand why certain optimizers are chosen for specific tasks in deep learning.

**Strengths:**

- I like that more recent neural network architectures such as  forward-forward neural network are considered
- In general the view of neural network training from the perspective of control and the respective methodology (like PID and fuzzy controllers) is of interest
- In general the paper tries to adress different domains, such as GAN-like training, vision, and feedforward neural networks

**Weaknesses:**

- The paper is written very poorly. It is very difficult to understand what the intentions of the the authors are. There are countless examples, starting with the title: "Based on What We Can Control Artificial Neural Networks"   Other examples include for instance "To analyze the learning progress of most ANNs, for example, CNN using backpropagation algorithm, FFNN using forward-forward algorithm, and GAN such a generative model using random noise to generate sample" (page 2)

- I don't see how the experiments make sense given the research question. It look to me that standard classification tasks are considered, with standard architectures and evaluations.

**Questions:**

Could you answer in simple terms:
1. What are the contributions of this paper?
2. Why is the perspective of "optimizers are controllers" relevant?
3. What are the research questions you are trying to answer with the experiments?

---

### Meta-Review · Area_Chair_ZD8W · 2023-12-13

**Metareview:**

This paper proposes to consider the optimization of neural network as a control problem. Using this framework, the behavior of different optimizers is analyzed w.r.t. different network architectures.

All the reviewers agree that the manuscript is currently severely lacking. I here summarize the main concerns:
- The experimental evaluation does not fully support the claims of the manuscript.
- The novelty of analyzing optimization as a control problem is limited, and more connections to existing literature should be drawn.
- The text is generally difficult to read and understand.

Overall, I agree with the evaluation of the reviewers that the manuscript is not ready for publication.

**Justification For Why Not Higher Score:**

This manuscript is clearly not ready for publication.

**Justification For Why Not Lower Score:**

N/A

---

### Decision · Program_Chairs · 2024-01-16

Reject